# Is quality maternal healthcare all about successful childbirth? Views of mothers in the Wa Municipality, Ghana

Linus Baatiema[1]ᴼ*, Augustine Tanle[2]‡, Eugene Kofuor Maafo Darteh[2]‡, Edward Kwabena Ameyaw[3]‡

1 Ghana Health Service, Upper West Regional Health Directorate, Wa, Ghana, 2 Department of Population and Health, College of Humanities and Legal Studies, University of Cape Coast, Cape Coast, Ghana, 3 Faculty of Health, The Australian Centre for Public and Population Health Research, University of Technology Sydney, Sydney, Australia

ᴼ These authors contributed equally to this work.
‡ These authors also contributed equally to this work.
* baatiemalinus@gmail.com

**Data Availability Statement:** The data underlying the results presented in the study have been uploaded as a supporting file.

## Abstract

### Introduction

In spite of the countless initiatives of the Ghana government to improve the quality of maternal healthcare, Upper West Region still records poor childbirth outcomes. This study, therefore, explored women's perception of the quality of maternal healthcare they receive in the Wa Municipality of the Upper West Region of Ghana.

### Materials and methods

This is a qualitative cross-sectional study of 62 women who accessed maternal healthcare in the Wa Municipality of Ghana. We analysed the transcripts using the analytic inductive technique. An inter-coding technique (testing for inter-coding agreement) was employed. The iterative coding process resulted in a coding scheme with four main themes. We used peer-debriefing technique in ensuring credibility and trustworthiness.

### Results

Logistics and equipment; referral service; empathic service delivery; inadequacy of care providers; affordability of service; satisfaction with services received; as well as experience and service delivery were the parameters used by the women in assessing quality maternity care. A number of gaps were reported in the healthcare system including limited healthcare providers, limited beds and inefficient referral system. Conversely, some of them reported that some healthcare providers offered empathetic healthcare. Contrary views were expressed with respect to satisfaction with maternity care.

### Conclusion

Government and all stakeholders seeking to enhance quality of maternal health and accelerate the attainment of the third Sustainable Development Goal need to reconsider the

**Funding:** The author(s) received no specific funding for this work.

**Competing interests:** The authors have declared that no competing interests exist.

financing of service delivery at health institutions. Indeed, our findings have illustrated that routine workshops on empathetic healthcare are required in efforts to increase the rate of facility-based childbirth, and thereby subside maternal mortality and all adverse pregnancy outcomes.

## Introduction

Poor healthcare during childbirth is more prevalent in low- and middle-income countries where most women do not obtain an acceptable level of care [1–3]. Higher than predicted maternal mortality occur in health facilities in lower-income countries despite the availability of required medical supplies in some instances, suggesting clinical management gaps or treatment delays for women who develop obstetric complications [4,5]. Low healthcare quality can compromise Ghana's prospects of achieving the Sustainable Development Goal (SDG) 3 target 3.8, which requires that everyone should have access to affordable and quality healthcare [6].

Quality maternal healthcare has been a policy and public health concern in sub-Saharan Africa (SSA) as the region disproportionately bears a high proportion of the global maternal mortality burden [7]. Despite major progress towards reducing maternal mortality over the last decade, many countries in SSA, including Ghana, were unable to achieve the MDG target of 75% decline in maternal mortality [8], partly attributable to the quality of healthcare. Meanwhile, childbirth in a well-resourced health facility with effective healthcare guarantee timely management and treatment of complications in order to reduce the chances of maternal and newborn deaths [9].

Several studies have examined the quality of childbirth care in Ghana [10–12] as well as maternal and neonatal care [13–15]. Ineffective functioning of existing administrative structures, inadequacy of equipment, limited commodities and registries, non-adherence to laboratory examination and unprofessional attitude of health workforce are some of the problems found [16–18]. Meanwhile attaining quality healthcare delivery in Ghana revolves around a complex interaction of economic, financial, social, and cultural factors that affect access to service and its quality [19].

More than 96% of pregnant women aged 15–49 years receive antenatal care in Ghana [20]. However, institutional childbirth is relatively lower (68%), despite the free maternal healthcare policy [21]. An assessment suggested that though Ghana's maternal healthcare policy has led to an increase in institutional childbirth, maternal mortality (310/100000) still remains high [21]. The relatively lower institutional childbirth in Ghana intensifies the chances of adverse childbirth complications because skilled birth assistants are more probable to be missed [22]. It is well established that poor quality of healthcare dissuade women from visiting health facilities for childbirth [23–25].

In spite of the several nationwide initiatives such as the free maternal healthcare policy, the Ghana National Health Insurance Scheme and the Safe Motherhood Initiative among others [26,27], the Wa Municipality is the worst performing district with regards to maternal mortality in the Upper West region. For instance, from 2015 to 2018, maternal mortality ratios were 335.3, 308.9, 2532, 152.2 and 252.2 per 100,000 live births respectively. Although the ratio shows a decreasing trend, they are still unacceptably high relative to the global target of 70 maternal deaths per 100,000 by 2030 [28]. Conversely, best performing districts like Nadowli-Kaleo and Wa West had 43.8 and 45.9 MMR in 2020 respectively. This makes the Wa Municipality a typical case requiring a scientific inquiry to understand the systemic factors leading to poor childbirth outcomes.

Severe bleeding, anemia, and other delivery complications are reported as the leading causes of maternal mortality in the region [29]. These causes are attributable to healthcare provision factors and therefore warrant the need for an investigation into the quality of maternal healthcare rendered to women at birth in the Wa Municipality. To date, empirical study on quality maternal healthcare, in the Wa Municipality, from mothers' perspective is non-exist. This study, therefore, seeks to address this apparent gap by exploring the account of women who have experienced the service in order to provide evidence to underpin policy efforts required to mitigate institutional level drivers of adverse childbirth outcomes.

## Theoretical framework

The Donabedian [30] model for measuring quality healthcare has been adapted as the theoretical framework to guide the study. The model offers a suitable lens through which quality maternal healthcare at childbirth can be comprehended. The model has three main blocks: structure, process and outcome. Structure refers to the characteristics of the setting (health facility) in which care takes place and extends to the care provider (e.g. midwives, policies and models of care). Donabedian [30] conceived 'process' as assessing whether a patient received what is acceptable as good care. Process encompasses any service that is rendered as part of the interaction between a healthcare professional and the patient, including interpersonal processes, such as providing information and emotional support, as well as involving patients in decisions in a way that is consistent with their preferences. Lastly, outcomes imply a patient's health status or change in condition (e.g. successful childbirth) resulting from the care received. Some literature on quality maternal healthcare have also posited that, for outcomes to be considered as quality indicators, they must reflect, or be responsive to, variations in the care assessed [31,32]. These three components shape a woman's interpretation of facility based childbirth experience. The framework was extremely beneficial as it offered critical directions in the conduct of the study and interpretation of the study results.

## Materials and methods

**Study design.** The study followed a cross-sectional study design. Cross-sectional design was expedient because data were collected from the study population once within a specific period (from May to June 2018).

**Setting and participants.** The study took place in the Wa Municipality, which is the capital of the Upper West Region of Ghana. The Municipality has a total population of 107,214 comprising 54,218 (50.6%) females and 52,996 (49.4%) males. About 54.8 percent of the population aged 15 years and older are economically active whilst 45.2 percent are economically inactive [33]. Of the economically active population, 91.5 percent are employed while 8.5 percent are unemployed. Agricultural production in Wa Municipal is chiefly concerned with the production of yam, which accounted for 69.8 percent of the total agricultural production during 2010–2015. A total of 38 health facilities were present in the district at the time of the study [34]. The study targeted women within the reproductive age group (15–49) who had given birth at the various health facilities in the Municipality. The inclusion criteria were that the woman should be a resident of the Municipality and should have accessed skilled delivery care at any of the health facilities within the Municipality within the last three years. The three-year period was deemed suitable in order to reduce recall biases with respect to their experiences and assessment of the care received. The study excluded women who sought ANC and caregivers/husbands who accompanied women to the facility for childbirth.

**Sampling and data collection.** A total of 14 health facilities were purposively selected for the study. These are CHPS zones (Boli CHPS Zone, Kpongu CHPS Zone, Dondoli CHPS

Zone, Sombo CHPS Zone, Kperisi CHPS Zone, Dobile CHPS Zone, Beehe CHPS Zone, Piisi CHPS Zone, Busa Health Centre), Health Centres (Charia Health Centre, Kambali Health CHPS Zone, Charingu CHPS Zone, Wa Urban Health Centre) and lastly Regional Hospital. For a health facility to be included, it should have at least a midwife or a doctor who attend to childbirth. The study, however, utilised convenience sampling technique to recruit the women. These women were recruited during their postnatal care (PNC) visits to the 14 health facilities. With this approach, women who attended PNC and were exiting the health facility were approached and after giving consent, were interviewed. The whole process was repeated until the study got to saturation with a sample of 62. The study carefully recruited this large sample to overcome some of the possible biases associated with convenience sampling.

**Data collection instruments.** The study utilised two research instruments. The tools utilised are in-depth interview (IDI) guide and Focus Group Discussion (FGD) guide. These guides were developed to reflect the underlying objective of the study.

**Data collection procedure.** Face-to-face in-depth interviews and focus group discussions were conducted with the participants. All the respondents were informed of the objectives of the research and freely decided to participant or decline. Prompts and follow up questions were frequently used to gain insights into the responses and to clarify responses from the interviewees. In-depth interviews were conducted according to the date, time and place agreed upon with each mother. Interviews mostly occurred at either their homes or work places. Irrespective of location, interviews were conducted without the presence of a third party. All interviews were conducted in the local dialects; Dagaare, Waale, and Brefo and lasted between 25–35 minutes on average.

Focus group discussions (FGDs) were conducted to consolidate findings from the in-depth interviews. Women who presented typical or extreme cases were selected for the FGDs. Extreme cases involved most pathetic situations shared during the FGD. For instance, a near miss situation or loss of fetus in the process of care. On the other hand, typical cases referred to normal maternity conditions such as normal childbirth. Some women who met these conditions were recruited for the IDIs. This was done to be sure if their responses were not influenced by the presence of other women (during the FGD). In order to have uninterrupted conversations, we held discussions with only mothers, and meetings were held outside a church or on a community member's compound. One FGD was conducted in each of the six sub-districts of the Municipality. Participants were mothers who had delivered in a health facility. FGDs were conducted in the local language, moderated by one interviewer (LB or a research assistant) whilst one research assistant took notes and audio recorded the discussion. Each FGD lasted approximately 30–45 minutes. The membership of each FGD session ranged from 6 to 9.

**Pre-testing of instrument.** As part of quality control checks, we pre-tested the tools in non-study sites among six CHPS zones, one mother each who had delivered in the health facility. The pre-test was conducted to check the suitability of sequencing and framing of questions and whether all questions were appropriate within the study context [35,36]. The outcome from the pre-test aided in refining the instrument prior to the actual data collection exercise.

**Data analysis.** The analysis began with verbatim transcription of audio files and translated into the English Language afterwards. This was followed by manual coding of transcripts. We conducted analysis of the transcripts using the analytic inductive technique [37]. This was achieved by condensing the raw data into categories or themes based on the literature review and the theoretical framework underpinning the study. The second stage involved immersion into the data by allowing the themes to emerge from the data [38]. To check the consistency of the themes/codes, an inter-coding technique (testing for inter-coding agreement) was used to

test the codes. The iterative coding process resulted in a coding scheme with four main categories or themes.

**Credibility and trustworthiness.** To ensure credibility and trustworthiness, peer-debriefing technique was used. The peer debriefing or member check technique was further used to validate the interpretations given to some of the codes and quotations [39]. Here, the codes together with the interpretations were given to AT and EKMD to check for consistency in quotes and thoughts regarding the interpretations and compare them to the ones originally espoused. Where some inconsistencies were raised in terms of the interpretation, appropriate adjustments were done by contacting the women for clarity. The results of this exercise aided in ensuring that codes, quotes, and interpretations largely represented what the women experienced and communicated. The use of these techniques greatly improved on the trustworthiness, dependability, consistency and transferability of the results reported in this study.

**Ethics.** The study received approval from the Institutional Review Board (IRB) of the University of Cape Coast (UCCIRB/CHLS/2018/05). Prior to each interview, written consent was obtained from participants for the interview and the audio recordings. Those who could not write thumb printed. For women under 18 years, permission was sought from their husbands, parents or caregivers. Those who agreed signed/thumb printed a consent form before being interviewed.

## Results

### Socio-demographic characteristics of respondents

Sixty-two mothers were interviewed for the study. The results indicated that 27 mothers were within the age group of 25–29 years (Table 1). Fifty-one of the service users were married. Twenty-six (26) had attained no formal education while 23 of the users had three children prior to the study. Twenty (20) of the mothers had their children born and they were alive. Twenty-six of the mothers were into Artisanship (hairdresser, craft woman, weaver, etc). Eleven (11) of the mothers were from Charia Health Centre.

### Appraisal of quality of maternal healthcare

The following are the thematic areas that were appraised by the research participants: logistics and equipment, referral service, empathic service delivery by providers, inadequacy of care providers, affordability of service, satisfaction with service received as well as experience and service delivery.

### Logistics and equipment

Most of the participants considered health care related logistics and equipment in the hospitals they visited as being inadequate, especially beds. They also added that the facility found at their CHPS cannot accommodate two or more deliveries due to the limited number of beds. Some women have to wait for the care providers to attend to other women before they could have access to beds. These are some assertions made by mothers:

*The rooms are not spacious enough. The day I went to deliver, four of us gave birth that day. We were sitting down waiting for one to finish before the other because the bed is only one. When one delivers then they start to supervise the next person* (30 years old; Unemployed from Sombo).

**Table 1. Socio-demographic background of the respondents.**

| Socio-demographic Characteristics | Frequency |
| --- | --- |
| | *f* |
| **Age** | |
| 20–24 | 8 |
| 25–29 | 27 |
| 30–34 | 20 |
| 35–39 | 7 |
| **Marital Status** | |
| Never Married | 11 |
| Married | 51 |
| **Level of Education** | |
| No Education | 26 |
| Primary | 13 |
| JHS | 11 |
| SHS | 3 |
| Tertiary | 9 |
| **Number of Children Born** | |
| 1 | 10 |
| 2 | 12 |
| 3 | 23 |
| 4 | 9 |
| 5 And Above | 8 |
| **Number of Children Alive** | |
| 1 | 11 |
| 2 | 14 |
| 3 | 21 |
| 4 | 9 |
| 5 and Above | 7 |
| **Occupation** | |
| Farmer | 17 |
| Trader | 11 |
| Artisan | 26 |
| Public Servant | 5 |
| Unemployed | 3 |
| **Sub-district** | |
| Busa | 9 |
| Charia | 11 |
| Charigu | 10 |
| Kambali | 10 |
| Wa Urban Health Centre | 5 |
| Bamahu | 10 |
| Regional Hospital* | 7 |
| Total | 62 |

Source: Field work, 2018 ||

* = Not a sub-district.

## Referral service

Due to limited resources at most health facilities that render maternal services, referral to higher facilities is a common phenomenon in maternal healthcare delivery. As such, some key informants and mothers shared their thoughts and experiences by way of appraising the ongoing referral services they experience in their communities. A discussant in a Focus Group Discussion, indicated that:

*Usually, when we are referred to the Wa Municipal Hospital from the Kpongu Clinic, the midwives sometimes tell us that the babies are not positioned well so we need to go to the bigger facility. However, from our experiences, there was really not much difference in the experience between the Clinic here and there* (Respondent 1, Trader, FGD at Kpongu).

Other woman described the entire referral process as being a problem due to the transportation arrangement, poor roads and the financial commitment involved. The women expressed themselves as follows:

*The major difficulty we encounter is the referral when we are given referral to Wa, it becomes difficult for us to get there. We have to hire a car. I was pregnant and water was running out of my genitals, I thought I was due to give birth, so I went to the health center. They examined me and realized that the child was not closer, yet the water was flowing, so they had to refer me to the Regional Hospital* (33 years old; Farmer from Boli).

Another participant remarked that:

*I could also die in the process of going to the Regional Hospital to deliver because the road from Charia connecting Wa is rough and full of potholes . . .. the kind of balancing this Motor King (tricycle) ridder had to take me through could have killed me . . . hmmm but do I really have a choice?* (34 years; Public Servant from Charia).

## Empathic service delivery

It was realized that a number of the participants were touched by the diligence and empathy some healthcare providers attached to the maternal health services they rendered to them. These are some of the thoughts shared by the participants:

*The nurses are reliable, anytime you call her she comes around to attend to you. They don't sleep here but they are responsive. For the midwife when she even closes and goes home and you call her to come and perform deliveries she comes back* (30 years old; Unemployed from Kpongu).

*Hmmmmmmm. . ..is not easy ooooo, these inexperienced personnel will kill us. Whenever we recommend for the big men to release experienced midwives to the labour ward, all they do is to gather these younger ones there. In fact, my last experience was bad when I came to deliver my second born. They could not detach the placenta, the whole process was like Caesarean Section without anesthesia or diazepam* (30 years old; Unemployed from Sombo).

Similarly, another discussant indicated that:

*The midwives are very timely to our call because when a nurse is to deliver you, she sits by you throughout especially at night to constantly check and take care of you. Even when the nurses*

*go to sit at their table and you call them, they are quick to come to your aid* (Respondent 1, Artisan, FGD at Kpongu).

On this, one woman commented:

"*the nurses are very empathetic to clients because they do not shout or treat clients badly. They try to share your pain with you; they are patient with you in the pain you go through till you deliver"* (29 years old; Trader from Sombo).

In spite of these, some of the participants had a contrary perspective to the diligence and empathic conduct of the care providers, particularly the nurses. On this, some women expressed their views as:

*Not all the nurses are reliable and responsive to client because when I went to deliver, I had a cut and when one of the young nurses who conducted the delivery saw that the cut was beyond her, she went out to call one of the elderly nurses who was sitting at the table (a senior nurse) to come and assist her, all she said was "I am not ready to stitch anything today". I had to lie down and wait for her before she came at her own convenient time to stitch (Respondent 2, Artisan, FGD at Dondoli).*

One of the mothers also commented:

*The nurses are very empathetic but there are times that the attitudes of "we" the women in labour force the nurses to shout at us in order for us to be serious to push. Because when at a point during labour the woman tries to complicate things that could lead to the death of the baby, they lose the empathy and discipline you (Respondent 6, Public Servant, FGD at Kpongu).*

One of the mothers shared her lamentation:

*At the Regional Hospital, nurses harass people. I saw women who were being harassed when we went first. Some of my colleagues who got pregnant whilst in school, they were insulting them and even refused to use the Dettol they brought to the health facility. They told them that the Dettol was very old. (32 old years; Farmer from Kpongu).*

## Inadequacy of care providers

One other essential indicator used by the research participants in appraising the quality of maternal healthcare they received was the adequacy of care providers. It was noted from the conversations with the participants that some of the health facilities have limited staff which sometimes compromises the quality of the services they received as a mother described in the following words:

*The personnel are not really much they are about five (5) but they are doing well, they try all their best to take care of us. . .there is only one midwife here and she does deliveries with the help of a Community Health Nurse here. (33 years old; Public Servant from Charia).*

Meanwhile, one participant had a contrary view and commented as follows: ". . .*and there are always a good number of nurses and midwives in the ward to take care of us".* (Respondent 1, Trader, FGD at Kpongu). Additionally, one mother noted that there are adequate healthcare providers to cater for maternal health services:

*There are enough personnel here and also the tools and equipment for delivery are enough and functioning well. There are two (2) midwives here who conduct delivery and they are very reliable and responsive to clients because immediately you get here, they come to you and begin to interact with you till you deliver.* (29 years old; Trader from Sombo)

## Affordability of service

The affordability of health service is another critical issue as far as the quality of maternal healthcare is concerned. Quite a number of the participants claimed that the care they received from the care providers was affordable:

*Yes, it is affordable for me because I paid Gh¢ 6 ($1.10). I was even lucky because most of the things I could not buy and the nurses told me to pay for and later dashed to me. I did not pay before I was discharged* (Respondent 4, FGD at Dondoli).

One of the mothers from Sombo also had this to share:

*We only pay Gh¢ 2.00 ($0.37) for light bill anytime we deliver at the facility. Everyone needs good and quality healthcare services after delivery. So if someone asks me to choose a health center for delivery, I will recommend this CHPS compound to the person* (31 years old; Artisan, Sombo).

Another key informant noted that the maternal care services offered to the women were affordable in his opinion due to the National Health Insurance "*It is very affordable because health insurance covers all. The only difference is the few things that you buy".* (29 years old; Public Servant from Charia).

## Satisfaction with services received

As to whether the service users were satisfied with the various services received by the care providers or not, it was realized that whilst some were satisfied with the service, others were not. Some of the reasons advanced for satisfaction with services received from care providers were as follows:

*Yes, I am very satisfied with the overall maternal healthcare services received here because the nurses care and are very patient with clients and also do their best to help you deliver safely and take your child in good health. The processes involved when I went to deliver were that they took my card and maternity book, checked my BP and items I brought (Dettol, parazone, rags and others). I was given a drip and asked to walk around until I felt like going to the toilet and when the nurses checked me up at that time they said I was due, I was then delivered, the placenta was also taken out and they cleaned me and my baby* (30 years old; Public Servant from Kperisi).

One of the mothers from Charingu also had this to share:

*I am very satisfied with the maternal healthcare services in the facility. When you are due for delivery, they ask you to bring Dettol parasol, soap, rubber and pads. During the delivery, they normally ask you to lie on the bed and they will put hand gloves on their hands and insert their fingers into your vagina to find out whether the child is closer or not. When you cannot push by yourself, they normally provide a machine to help you push out the child.* (30 years old; Farmer, Charingu).

*A trader in Dondoli added that*:

*Well, to me I am very satisfied with the conduct of nurses I met when I went to the facility for delivery. But because they are always changing, I am not sure if all of them are good but those I met when I went to deliver were very good. Right from the receiving point to deliver, there were good. They really assisted me left right (29 years old; Trader from Dondoli).*

However, some indicated that they were not satisfied with the maternal healthcare services rendered by the healthcare providers. Some of their complaints included the following:

*Am not very satisfied with the delivery process. A case where you go to the hospital to deliver and these "small small nurses", will sit and expect you who is in labour to spread the cloth and rubbers and be waiting for the baby to come. All they do is sit with their phone "kiki! kiri! kiri. . ." while you are suffering. . . . sometimes when they prescribe a drug for you to buy, they will ask you who is in labour to get to the pharmacy to get the drugs especially if you don't have a guardian (31 years old; Farmer from Wa Urban Health Centre).*

## Experience and service delivery

Quality of maternal healthcare was also appraised in light of one's experience. To a greater extent, they were of the view that the more experienced providers offered high-quality services than the novices in the service. This was summarized by one participant who further proposed that the experienced nurses ought to educate the younger ones;

*I also think the experienced nurses should educate the younger ones because the last time I went for one of the monthly check-ups, one of the young nurses wrote that I should go and take a scan and that was around my seventh month on pregnancy but when I got to the table one of the elderly and experienced nurses told me that scan is only done during the ninth month so I had to go and come back for them to rewrite the scan for the ninth month. So, I think the young nurses are not very experienced because they calculate wrongly for clients which I am not very happy about (Respondent 5, Public Servant, FGD at Dondoli).*

**Another mother out of anger has this to share**

*Hmmmmmmm. . ..is not easy ooooo, this inexperience I choose to call them will kill us but whenever we recommend during ANC for the big men to release experience midwives to labour ward, all they do is gather these younger once there. In fact, my last experience when I came to deliver my second born was very bad, the treatment I received from these nurses at that time was unbearable. My brother, after these midwives delivered me, they couldn't detach the placenta from me and the baby, the whole process was like CS without anesthesia or diaze- pam. I felt the pain and even after we were rescued by a doctor, I was bed redeem for two weeks (Respondent 2, Artisan from Regional Hospital).*

A Public Servant from Wa Urban Health Centre had this sad story to share:

*". . ..my brother is a pity, am a staff midwife who is on leave to delivery at my in-laws place. The manner and the way these young midwives were handing one woman next to the bed I was laying in labor ward got me burning. While the women were making much efforts to be pushing for the baby to come out, the midwife was just there watching these telenovelas the television. Out of anger just didn't know the time I got down from my bed to shout on her to*

*prioritized the woman in labour. Shortly after that, the woman delivered but was excessively bleeding and this same young midwife didn't even know she was supposed to provide a drug by injection or intravenous to stop uterine bleeding simple because she was terrified. There nooooo one of the cleaners mimed by saying that, these young midwives that's how their life's are in here, even we having seen anything yet.some out of confusion, they sometime mixing procedures to talk what you have seen and when you want to talk, is always like once you didn't sit in the midwifery classroom them, u don't have the power to advice their duty"* (Respondent 11 Public Servant from Wa Urban Health Centre).

## Discussion

Issues of maternal health have been a national concern whilst the provision of quality maternal healthcare has proven to be an approach for reducing maternal mortality ratios. Following this, the study explored the quality of maternal healthcare from the perspective of women who had experienced the service in the Wa Municipality of Ghana. There were reports of limited beds and space. Limited beds and space may compromise quality of service at the CHPS level in diverse ways considering the fact that the number of women who might be in need of any maternal services at any given time cannot be estimated with certainty. However, limited number of beds and spaces at CHPS facilities is a deliberate policy to prevent CHPS facilities shifting focus from preventive and outreach services to curative services. Thus, only emergency deliveries are encouraged at that the CHPS level, all other cases are required to be referred to facilities with the requisite beds and space.

As such, when more women's conditions necessitate beds, those who report later would have to wait. The limited bed and space may contribute to a third delay, that is, delay in receiving health care after reaching the facility [40]. The issue of limited logistics and equipment is not only a challenge in Ghana but in other places such as Nepal where Karkee, Lee and Pokhare [41] noted that women viewed public hospitals as low rated with regard to adequacy of rooms, water, environmental cleanliness, privacy and access to information [41]. When women access maternal healthcare from a particular facility and consistently experience a shortage in the required logistics and equipment, they could develop a negative perception about the facility [42]. The Social Theory of Perception explains the possible perceptions to be developed when people are satisfied with a service.

Some participants indicated that most care providers were empathic and diligent in discharging their duties. This is likely to induce positive perceptions among women. With this, the Social Theory of Perception argues that an expectant mother's first impression about healthcare influences her perception of service utilization in the future. This happens by either drawing service users unto the facility or forcing mothers to withdraw from the facility [42]. A positive experience at the health facility is highly esteemed by women as noted in Nigeria [43].

Donabedian [30] conceived that "process measures" explains whether a patient received what is known to be good care. However, if the inexperience of some care providers is too obvious such that care receivers can identify, that might affect the care receivers' interpretation of whether the service they received was good or not. The functionality of the inexperienced staff can be said to be below the expectation of the care receivers, which represents an imbalance in the view of the Functionalist Theory [44–46]. Healthcare providers are expected to render their services to meet the standard of care of a service user before a system can be considered as balanced. Again, in the conceptualization of Merton [44], Parsons [46] and Parsons [45], each social institution contributes important functions in order for society to thrive. As such, in this context, all healthcare providers offering services should have the required experience and skills to deliver services to the expectation of the care receivers.

A major limitation of our sampling approach is non-representativeness and high tendency of sampling error. To overcome this, we carefully recruited a very large sample of participants who fully fitted the inclusion criteria.

## Conclusion

The study revealed that women experience some positive attitudes or empathetic care with some exceptions. Generally, health was affordable and respondents were largely satisfied with the quality of care received though there were a few exceptions. Referrals systems were hampered by lack of transport services, poor roads and other delays. Beds, spaces and lack of personnel posed some challenges to quality maternal health care service delivery. Health training institutions need to empasise the importance of empathetic in order for all healthcare providers to upload this as an important virtue before they pass out from school to practice. Routine workshops on empathy and professionalism cannot be ruled out in efforts to increase facility delivery rate and overcome maternal mortality and all adverse delivery outcomes. To ensure effective referral service, there is the need to improvise readily available means of transport, such as tricycles, in order to overcome the common transportation challenges.

## Supporting information

**S1 Data.**
(DOCX)

## Acknowledgments

The authors are much grateful to the Wa Municipal Health Directorate and the service users within the municipality for the support as well as all their respondents.

## Author Contributions

**Conceptualization:** Linus Baatiema.

**Data curation:** Linus Baatiema, Edward Kwabena Ameyaw.

**Formal analysis:** Linus Baatiema, Edward Kwabena Ameyaw.

**Investigation:** Linus Baatiema, Eugene Kofuor Maafo Darteh, Edward Kwabena Ameyaw.

**Methodology:** Linus Baatiema.

**Project administration:** Augustine Tanle, Eugene Kofuor Maafo Darteh.

**Resources:** Linus Baatiema, Augustine Tanle, Eugene Kofuor Maafo Darteh, Edward Kwabena Ameyaw.

**Software:** Linus Baatiema, Augustine Tanle, Eugene Kofuor Maafo Darteh, Edward Kwabena Ameyaw.

**Supervision:** Augustine Tanle, Eugene Kofuor Maafo Darteh.

**Validation:** Augustine Tanle, Eugene Kofuor Maafo Darteh, Edward Kwabena Ameyaw.

**Writing – original draft:** Linus Baatiema.

**Writing – review & editing:** Linus Baatiema, Augustine Tanle, Eugene Kofuor Maafo Darteh, Edward Kwabena Ameyaw.

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
