## [Decision Letter · Decision Letter 0]

9 Apr 2021

PONE-D-21-05332

Is quality maternal healthcare all about successful childbirth? views of mothers in the Wa Municipality, Ghana

PLOS ONE

Dear Dr. Baatiema,

Thank you for submitting your manuscript to PLOS ONE. After careful consideration, we feel that it has merit but does not fully meet PLOS ONE’s publication criteria as it currently stands. Therefore, we invite you to submit a revised version of the manuscript that addresses the points raised during the review process.

The reviewers' comments both suggest a minor revision. Most of their comments deal with inputting further details and further explaining different concepts and methodology, which the authors should be particular attention to when re-reporting in a new version.

We look forward to receiving your revised manuscript.

Kind regards,

Michael Wells

Academic Editor

PLOS ONE

Journal Requirements:

2. When reporting the results of qualitative research, we suggest consulting the COREQ guidelines: http://intqhc.oxfordjournals.org/content/19/6/349. In this case, please consider including more information on the number of interviewers, their training and characteristics. Moreover, please provide the interview guide used as a Supplementary file.

Reviewers' comments:

Reviewer's Responses to Questions

**Comments to the Author**

1. Is the manuscript technically sound, and do the data support the conclusions?

Reviewer #1: Partly

Reviewer #2: Yes

2. Has the statistical analysis been performed appropriately and rigorously? 

Reviewer #1: N/A

Reviewer #2: Yes

3. Have the authors made all data underlying the findings in their manuscript fully available?

Reviewer #1: Yes

Reviewer #2: Yes

4. Is the manuscript presented in an intelligible fashion and written in standard English?

Reviewer #1: Yes

Reviewer #2: Yes

5. Review Comments to the Author

Reviewer #1: Minor Issues

1. Line 87, the reference for the maternal mortality figure is quite dated. Please use a more recent figure and reference.

2. Line 130-131, please reconcile the data collection period. You indicate May-June 2018 here, but the table source indicates Field Work, 2019.

3. Line 141, please list the 14 health facilities used as sites for recruiting participants. Will it be possible to indicate their status, for instance which of them were CHPS compounds, Health Centers, Polyclinics, Hospitals etc.?

4. Line 165, kindly explain what is meant by ‘typical or extreme cases’

5. Line 212, please check the spelling of weaver, it appears as ‘waiver’

6. Line 212-213, please clarify the use of Wa Urban health Center (Regional Hospital), was it one of the 14 sites (facilities) for recruitment of respondents or a place of residence for some of the respondents. If the latter is the case, I suggest you use Kabanye, which the closest residential community to that facility.

7. Kindly clarify place of residents used in Table 1. Were they the sites of health facilities used to recruit respondents or actual residential locations of the respondents? In the responses, other communities such as Sombo, Dondoli, Boli and Kpongu are mentioned. But the total number for the six locations mentioned in the table sum up to 62. Please check and make it clear.

8. Lines 255-259, the verbatim quote seems to be describing an experience which occurred in or around Charia, and yet it is attributed to a ‘public servant from Wa Urban Health Center’, please explain or rectify the anomaly.

Major Issues

9. From lines 394-401, authors allude to logistics and equipment constraints without mentioning a single equipment or the type of logistics involved. The data from the respondents only mention limited beds and spaces. I suggest you put this in context, the type and nature of equipment will vary from one facility to the other. It will be prudent to situate the discussion in its appropriate context. The limited number of beds and spaces at CHPS facilities is a deliberate policy to prevent CHPS facilities shifting focus from preventive and outreach services to curative services. Only emergency deliveries are encouraged at that the CHPS level, all other cases are required to be referred to facilities with the requisite staff and logistics to attend to the clients.

10. Similarly, the issue of staffing needs to be put in the right context as well. CHPS staffing norms requires a minimum of two and at best three health workers with different skills mix. Other higher facilities have different staffing norms. Which facilities did the respondents complain about lack of staff? The results and discussions do not make this clear.

11. The issue of (in)experience of some nursing staff have not been properly articulated. The results presents only one verbatim quote alluding to a possible incompetent handling of a scan request. This is not enough to draw conclusions on such an issue. Further and better particulars will be required.

12. Lines 426-433, the conclusion needs to be rewritten. The current conclusion does not reflect the findings of the study. It does not appear to drive home the main message of the study, which to me are (1) respondents reported some positive attitudes (empathy) of health workers with some exceptions; (2) health was affordable and most cost effective; (3) respondents were largely satisfied with the quality of care received, with a few exceptions; (4) referrals systems was still hampered by challenges of lack of transport services, poor roads and other delays; (5) Issues of beds, spaces and lack of personnel still posed some challenges to quality maternal health care service delivery.

13. No recommendations appear in the main article.

Reviewer #2: Is quality maternal healthcare all about successful childbirth? views of mothers in the Wa Municipality, Ghana

Dear editor,

Thanks for inviting me as reviewer on the above titled manuscript for PLoS ONE.

First, the authors study a topic that is relevant for achieving the Substianable Development Goals 3 in one of the most underdeveloped regions of Ghana.

I will recommend that the manuscript is accepted with minor revisions. Below are my suggestions for the authors to consider in their revision.

1. The authors provide an overview of the maternal mortality ratio in the Wa municipality in the Upper West Region of Ghana. The authors state that the municipality is the worst performing in the region in terms of MMR. I will suggest that the authors provide the MMR statistics for the best performing municipality (or average of the region) to provide context to the readers of the situation of the municipality. Such a comparison will provide an additional justification for studying quality maternal healthcare in the municipality.

2. The authors should also provide a brief socioeconomic background of the Wa municipality (or the Upper West Region).

3. Does the choice of a convenient sampling procedure affect the results of the study? How do the authors address the limitations of the sampling method to reduce bias in the findings of the study?

6. PLOS authors have the option to publish the peer review history of their article (what does this mean?). If published, this will include your full peer review and any attached files.

Reviewer #1: No

Reviewer #2: No

---

## [Author Response · Author response to Decision Letter 0]

24 Jul 2021

AUTHORS’ RESPONSE TO REVIEWS

Title: Is quality maternal healthcare all about successful childbirth? views of mothers in the Wa Municipality, Ghana

Manuscript ID: PONE-D-21-05332 

Dear Editor and Reviewer (s),

On behalf of all authors, I convey our gratitude to you for the critical and constructive review that has led to the improvement of our paper entitled “Is quality maternal healthcare all about successful childbirth? views of mothers in the Wa Municipality, Ghana”. We have revised the manuscript based on the comments raised. In the following detailed responses, we address each comment calling for changes point-by-point, indicating where relevant additional texts have been added to the body of the manuscript. We believe the manuscript has improved substantively and will be published in your reputable journal. 

Editor(s)' Comments to Author:

1. The authors provide an overview of the maternal mortality ratio in the Wa municipality in the Upper West Region of Ghana. The authors state that the municipality is the worst performing in the region in terms of MMR. I will suggest that the authors provide the MMR statistics for the best performing municipality (or average of the region) to provide context to the readers of the situation of the municipality. Such a comparison will provide an additional justification for studying quality maternal healthcare in the municipality.

Responses: Thank you. We have now provided statistics for best performing districts. Apart from Wa Municipality, all the other administrative regions are currently in the category of districts but not municipality.

2. The authors should also provide a brief socioeconomic background of the Wa municipality (or the Upper West Region).

Responses: This is done, thank you.

3. Does the choice of a convenient sampling procedure affect the results of the study? How do the authors address the limitations of the sampling method to reduce bias in the findings of the study?

Responses: We have indicated this in the methods and discussion sections.

Reviewer(s)' 1 Comments to Author:

1. Comment: Line 87, the reference for the maternal mortality figure is quite dated. Please use a more recent figure and reference

Responses: We have updated the reference as suggested. 

2. Comment: Line 130-131, please reconcile the data collection period. You indicate May-June 2018 here, but the table source indicates Field Work, 2019.

Responses: Many thanks for valid observation. We have reconciled this.

3. Comment: Line 141, please list the 14 health facilities used as sites for recruiting participants. Will it be possible to indicate their status, for instance which of them were CHPS compounds, Health Centers, Polyclinics, Hospitals etc.?

Responses: We really appreciate this. We have revised the listed all the 14 facilities and their classified them according to their statuses. Thus, clarity has also been established on whether these facilities were CHPS zones, Health centers or Regional Hospital. 

4. Comment: Line 165, kindly explain what is meant by ‘typical or extreme cases’

Responses: Extreme cases involved most pathetic situations shared during the FGD. For instance, loss of fetus in the process of care or a near miss situation. On the other hand, typical cases referred to normal maternity conditions such as normal childbirth. Some women who met these conditions were recruited for the IDIs. This was done to be sure if their responses were not influenced by the presence of other women (during the FGD).

5. Comments: Line 212, please check the spelling of weaver, it appears as ‘waiver’

Responses: We have corrected this. 

6. Comments: Line 212-213, please clarify the use of Wa Urban health Center (Regional Hospital), was it one of the 14 sites (facilities) for recruitment of respondents or a place of residence for some of the respondents. If the latter is the case, I suggest you use Kabanye, which the closest residential community to that facility

Responses: Thank you. Please we have clarified this. 

7. Comments: Kindly clarify place of residents used in Table 1. Were they the sites of health facilities used to recruit respondents or actual residential locations of the respondents? In the responses, other communities such as Sombo, Dondoli, Boli and Kpongu are mentioned. But the total number for the six locations mentioned in the table sum up to 62. Please check and make it clear.

Responses: Thank you very much for the comments. It is not their places of residence so we have revised to read, as “sub-districts”. Also, the 62 is the of total number of participants sampled from the six sub-districts (facilities) for the study.

8. Comments: Lines 255-259, the verbatim quote seems to be describing an experience which occurred in or around Charia, and yet it is attributed to a ‘public servant from Wa Urban Health Center’, please explain or rectify the anomaly.

Responses: The inconsistency in place of residence of the respondent is rectified now. See page

9. From lines 394-401, authors allude to logistics and equipment constraints without mentioning a single equipment or the type of logistics involved. The data from the respondents only mention limited beds and spaces. I suggest you put this in context, the type and nature of equipment will vary from one facility to the other. It will be prudent to situate the discussion in its appropriate context. The limited number of beds and spaces at CHPS facilities is a deliberate policy to prevent CHPS facilities shifting focus from preventive and outreach services to curative services. Only emergency deliveries are encouraged at that the CHPS level, all other cases are required to be referred to facilities with the requisite staff and logistics to attend to the clients

Responses: Thank you. We have revised and contextualized this. 

10. Similarly, the issue of staffing needs to be put in the right context as well. CHPS staffing norms requires a minimum of two and at best three health workers with different skills mix. Other higher facilities have different staffing norms. Which facilities did the respondents complain about lack of staff? The results and discussions do not make this clear.

Responses: We have clarified this now, thank you.

11. The issue of (in)experience of some nursing staff have not been properly articulated. The results present only one verbatim quote alluding to a possible incompetent handling of a scan request. This is not enough to draw conclusions on such an issue. Further and better particulars will be required.

Responses: Thank you. Please we have added another quote to buttress this point.

12. Lines 426-433, the conclusion needs to be rewritten. The current conclusion does not reflect the findings of the study. It does not appear to drive home the main message of the study, which to me are (1) respondents reported some positive attitudes (empathy) of health workers with some exceptions; (2) health was affordable and most cost effective; (3) respondents were largely satisfied with the quality of care received, with a few exceptions; (4) referrals systems was still hampered by challenges of lack of transport services, poor roads and other delays; (5) Issues of beds, spaces and lack of personnel still posed some challenges to quality maternal health care service delivery.

Responses: The conclusion has been rewritten to reflect the findings of the study. 

13. No recommendations appear in the main article.

Responses: Recommendations have been incorporated into the revised conclusion. 

REVIEWER: 2

1. Comment: The authors provide an overview of the maternal mortality ratio in the Wa municipality in the Upper West Region of Ghana. The authors state that the municipality is the worst performing in the region in terms of MMR. I will suggest that the authors provide the MMR statistics for the best performing municipality (or average of the region) to provide context to the readers of the situation of the municipality. Such a comparison will provide an additional justification for studying quality maternal healthcare in the municipality.

Responses: Thank you. We have now provided statistics for best performing districts. Apart from Wa Municipality, all the other administrative regions are currently in the category of districts but not municipality.

2. Comment: The authors should also provide a brief socioeconomic background of the Wa municipality (or the Upper West Region).

Responses: This is done, thank you.

3. Comment Does the choice of a convenient sampling procedure affect the results of the study? How do the authors address the limitations of the sampling method to reduce bias in the findings of the study?

Responses: We have indicated this in the methods and discussion sections.

---

## [Decision Letter · Decision Letter 1]

1 Sep 2021

Is quality maternal healthcare all about successful childbirth? views of mothers in the Wa Municipality, Ghana

PONE-D-21-05332R1

Dear Dr. Baatiema,

We’re pleased to inform you that your manuscript has been judged scientifically suitable for publication and will be formally accepted for publication once it meets all outstanding technical requirements.

Kind regards,

Michael Wells

Academic Editor

PLOS ONE

Additional Editor Comments (optional):

Both reviewers, as well as the editor have found that the authors have made substantial improvements that strengthen the quality of their manuscript and it is now ready for acceptance.

Reviewers' comments:

Reviewer's Responses to Questions

**Comments to the Author**

1. If the authors have adequately addressed your comments raised in a previous round of review and you feel that this manuscript is now acceptable for publication, you may indicate that here to bypass the “Comments to the Author” section, enter your conflict of interest statement in the “Confidential to Editor” section, and submit your "Accept" recommendation.

Reviewer #2: All comments have been addressed

2. Is the manuscript technically sound, and do the data support the conclusions?

Reviewer #2: Yes

3. Has the statistical analysis been performed appropriately and rigorously? 

Reviewer #2: Yes

4. Have the authors made all data underlying the findings in their manuscript fully available?

Reviewer #2: Yes

5. Is the manuscript presented in an intelligible fashion and written in standard English?

Reviewer #2: Yes

6. Review Comments to the Author

Reviewer #2: The authors have sufficiently addressed the comments raised in my previous review. The inclusion of MMR statistics from the best performing districts/municipalities in the Upper West region of Ghana highlights the importance of understanding the factors underlying such poor performance in the Wa municipality from a demand-side perspective.

7. PLOS authors have the option to publish the peer review history of their article (what does this mean?). If published, this will include your full peer review and any attached files.

Reviewer #2: No

---

## [Editor Report · Acceptance letter]

6 Sep 2021

PONE-D-21-05332R1 

Is quality maternal healthcare all about successful childbirth? views of mothers in the Wa Municipality, Ghana 

Dear Dr. Baatiema:

I'm pleased to inform you that your manuscript has been deemed suitable for publication in PLOS ONE. Congratulations! Your manuscript is now with our production department. 

Kind regards, 

on behalf of

Dr. Michael Wells 

Academic Editor

PLOS ONE